# Use of Capsid Integrity-qPCR for Detecting Viral Capsid Integrity in Wastewater

**DOI:** 10.3390/v16010040

**Published:** 2023-12-26

**Authors:** Jessica L. Kevill, Kata Farkas, Nicola Ridding, Nicholas Woodhall, Shelagh K. Malham, Davey L. Jones

**Affiliations:** 1School of Environmental and Natural Sciences, Bangor University, Bangor, Gwynedd LL57 2UW, UK; k.farkas@bangor.ac.uk (K.F.); n.ridding@bangor.ac.uk (N.R.); n.woodhall@bangor.ac.uk (N.W.); or davey.jones@murdoch.edu.au (D.L.J.); 2School of Ocean Sciences, Bangor University, Menai Bridge, Anglesey LL59 5AB, UK; s.malham@bangor.ac.uk; 3Centre for Sustainable Farming Systems, Food Futures Institute, Murdoch University, Murdoch, WA 6150, Australia

**Keywords:** human pathogenic viruses, One Health, WBE, WBS, viral inactivation, waterborne viruses, capsid integrity, viability qPCR, enteroviruses, influenza viruses, SARS-CoV-2, adenovirus

## Abstract

Quantifying viruses in wastewater via RT-qPCR provides total genomic data but does not indicate the virus capsid integrity or the potential risk for human infection. Assessing virus capsid integrity in sewage is important for wastewater-based surveillance, since discharged effluent may pose a public health hazard. While integrity assays using cell cultures can provide this information, they require specialised laboratories and expertise. One solution to overcome this limitation is the use of photo-reactive monoazide dyes (e.g., propidium monoazide [PMAxx]) in a capsid integrity-RT-qPCR assay (ci-RT-qPCR). In this study, we tested the efficiency of PMAxx dye at 50 μM and 100 μM concentrations on live and heat-inactivated model viruses commonly detected in wastewater, including adenovirus (AdV), hepatitis A (HAV), influenza A virus (IAV), and norovirus GI (NoV GI). The 100 μM PMAxx dye concentration effectively differentiated live from heat-inactivated viruses for all targets in buffer solution. This method was then applied to wastewater samples (*n* = 19) for the detection of encapsulated AdV, enterovirus (EV), HAV, IAV, influenza B virus (IBV), NoV GI, NoV GII, and SARS-CoV-2. Samples were negative for AdV, HAV, IAV, and IBV but positive for EV, NoV GI, NoV GII, and SARS-CoV-2. In the PMAxx-treated samples, EV, NoV GI, and NoV GII showed −0.52–1.15, 0.9–1.51, and 0.31–1.69 log reductions in capsid integrity, indicating a high degree of potentially infectious virus in wastewater. In contrast, SARS-CoV-2 was only detected using RT-qPCR but not after PMAxx treatment, indicating the absence of encapsulated and potentially infectious virus. In conclusion, this study demonstrates the utility of PMAxx dyes to evaluate capsid integrity across a diverse range of viruses commonly monitored in wastewater.

## 1. Introduction

The use of wastewater-based surveillance (WBS) to monitor community-level health via the detection of harmful disease-causing pathogens, excreted in faeces and urine, has greatly increased in recent years [1]. For example, many studies now exist showing the utility of WBS to detect viruses, such as norovirus [2], polio and non-polio enterovirus [3,4,5], adenovirus [6,7,8], hepatitis A [9,10], hepatitis E [11,12], and respiratory viruses such as coronaviruses, influenza viruses, and respiratory syncytial virus [13,14,15]. During the COVID-19 pandemic, WBS was widely implemented for population-level surveillance of SARS-CoV-2 by research groups and laboratories globally [16,17,18,19,20,21,22], further showcasing the usefulness of WBS for disease monitoring and public health policy implementation. 

Viruses in wastewater are typically detected by first concentrating and precipitating the viruses, extracting their nucleic acids, and then performing molecular techniques on the samples such as RT-qPCR (real-time quantitative polymerase chain reaction), dd-PCR (digital droplet PCR), or next generation sequencing. However, these methods cannot distinguish between intact potentially infectious viruses and degraded non-infectious viral particles; rather, they detect genetic markers from viruses regardless of viability. Thus, while the data gathered by these techniques provide an overview of infection levels in the community, it does not indicate the viral stability or the potential for infectivity if an individual comes into contact with sewage-contaminated water. This aspect of WBS is becoming more important, given the increased discharge of untreated sewage into rivers and seas and the associated risks to human and environmental health (i.e., One Health) [23,24]. 

Cell culture is the gold standard for virus viability testing. However, it requires specialised equipment like incubators, inverted microscopes, centrifuges, plate readers, hemacytometers, cryogenic storage, CO_2_, media, and other consumables [25]. In addition, it can take 4–7 days to generate infectivity data, causing delays in the implementation of public health interventions. It also requires staff with expertise to maintain the cell lines. Furthermore, many human respiratory viruses can only be cultured in biosecurity level (BLS) 3 laboratories, which are usually not available in environmental monitoring facilities. These requirements can be limiting factors, especially in lower- or middle-income countries (LMICs), where, in addition, frequent power outages may be common. Furthermore, isolating viruses from wastewater for cell culture assays is extremely challenging. Viruses are often present at low viral loads; so, large volumes (>10 l) of wastewater must be concentrated prior to culturing. Wastewater also contains debris and microbes that can contaminate and destroy cell cultures used for virus viability assays. Therefore, methods like filtration are needed to remove these contaminants without damaging viable viruses [26]. Antibiotics also may need to be added to the culture media to reduce microbial activity, albeit at levels that do not compromise the cell line [27]. Therefore, simpler methods like capsid integrity-PCR/qPCR (ci-PCR or ci-qPCR) (also referred to as viability qPCR/RT-qPCR) are needed to assess the potential infectivity of viruses in wastewater samples.

Capsid integrity-PCR/qPCR has become an important technique in virology for selectively detecting potentially infectious viral particles. Photo-reactive monoazide dyes such as ethidium monoazide (EMA), propidium monoazide (PMA), and PMAxx^®^ penetrate damaged viral capsids with exposed genomes, which are incapable of cell attachment and entry but are excluded from viruses with intact capsids that maybe capable of host cell infection. After addition of the dye to a wastewater viral concentrate, the sample is exposed to light at which point the dye modifies any exposed nucleic acids, blocking their subsequent amplification by PCR [28]. In contrast, the genomes of viruses with intact capsids that are potentially infectious can be successfully amplified. This allows selective quantification of the encapsulated viral genomes versus the non-infectious nucleic acids present in the sample. Capsid integrity-PCR using monoazide-based dyes has been applied to differentiate infectious and non-infectious viruses including influenza-A virus [29], norovirus [30], and SARS-CoV-2 [31]. Furthermore, the quantification of encapsulated viral genomes has provided insights into viral persistence in the environment [32,33] and the disinfection efficiency [34,35,36]. It is worth noting that the capsids of viruses inactivated using UV are still intact, and as a result, ci-qPCR is not suitable for assessing UV disinfection [36]. However, the sensitivity and specificity of this approach continues to be improved through optimising dye types, the concentration of dye used, incubation times, and light exposure length [37]. 

Many studies have utilised ci-qPCR to assess the capsid integrity of viruses in contaminated drinking water and food, as reviewed [38]; however, the use of ci-qPCR in wastewater monitoring has received much less attention. Of the wastewater studies that do exist, ci-qPCR has been used to assess the capsid integrity of adenovirus [39], enterovirus [39,40], norovirus GI and GII [30,32,40,41], rotavirus [32,39], hepatitis A virus [10,42], and SARS-CoV-2 [31,32]. Human enteric virus genomes (adenovirus, enterovirus, and norovirus) in wet and dry wastewater-derived struvite have also been shown to remain encapsulated using intercalating dyes [43]. We hypothesize that wastewater-based ci-qPCR can provide improved estimates of public health exposure risks compared to conventional PCR, especially regarding the potential infection risks posed by sewage discharges into the environment. This study therefore aims to optimise and assess the usefulness of ci-RT-qPCR using PMAxx dyes alongside conventional wastewater monitoring via RT-qPCR, for important viruses of public health concern including adenovirus (AdV), enterovirus (EV), hepatitis A (HAV), influenza A virus (IAV), influenza B virus (IBV), norovirus GI (NoV GI), norovirus GII (NoV GII), and SARS-CoV-2. The optimisation of PMAxx dye for a range of viral targets is also presented. 

## 2. Materials and Methods

### 2.1. Viral Stocks

The viral stocks used for PMAxx dye optimisation were AdV 40 cultured in-house, as described previously [44], HAV (Culture Collections of the UK Health Security Agency), IAV vaccine strain (H1N1) (kindly provided by Dr Eleanor Gaunt, University of Edinburgh), and NoV GI (Culture Collections of the UK Health Security Agency). To optimise the PMAxx dye assays on live and heat-inactivated viruses, samples were made containing AdV at concentration of ~10^8^–10^9^ gene copies/ml (gc/ml), HAV at ~10^5^–10^6^ gc/ml, IAV at ~10^7^–10^8^ gc/ml, and NoV GI at ~10^5^–10^6^ gc/ml. Viruses were suspended in phosphate buffer saline (PBS), pH 7.4, to a final volume of 5 ml. An aliquot of the virus mixture was then heat inactivated at 70 °C for 30 min in a benchtop heating block, to allow for a comparison of viruses with damaged capsids, alongside samples containing live viruses. All samples were made in triplicate.

### 2.2. PMAxx Optimisation 

PMAxx dye 20 mM in H_2_O (Biotium, Cat no: 40069; Biotium Inc., Freemont, CA, USA) was diluted with molecular grade H_2_O to a final volume of 10 mM to make a working stock. PMA enhancer at 1 X concentration was added to each sample and mixed by pipetting. Working in the dark, PMAxx dye working solution was added to the samples at a final concentration of 50 µM or 100 µM. Once the PMAxx dye was added, samples were incubated in the dark, at 20 °C on a bench top rocker (PMR-30 2D Rocker, Cat No: PMR-30-UK) at 30 oscillations/min. The samples were then exposed to light (465–475 nm) for 20 min at room temperature, using the PMA-Lite™ 2.0 Photolysis Device (Biotium, Cat No: BTE90006). Nucleic acids were extracted directly after light exposure. 

### 2.3. Wastewater Sample Collection

On 13 June 2023, 1 litre of influent wastewater samples was collected from behind the primary screen at 17 wastewater treatment plants (WWTP) and the main sewer line leaving 2 hospital sites, located in North Wales, resulting in a total of 19 samples. These sites formed part of the national wastewater-based public health surveillance programme funded by the Welsh Government. All wastewater samples were 24 h composites collected using refrigerated autosamplers, with samples taken every 15 min. The samples were collected and transported chilled to the laboratory, where on their arrival they were stored at 4 °C overnight and processed the following day. Sample storage at 4 °C is deemed appropriate for wastewater samples, as minimal impact on the viral decay and recovery of SARS-CoV-2 and surrogates from wastewater has been evidenced [45,46].

#### Wastewater Processing 

Samples were processed via the polyethylene glycol (PEG) precipitation method, as previously described [47,48] but with modifications [49]. Briefly, a PEG-NaCl solution at 40% PEG (PEG8000, Sigma-Aldrich, St. Louis, MO, USA, Cat. No. P5413) with 8% NaCl (Sigma-Aldrich, St. Louis, MO, USA, Cat. No. S7653) was prepared. Wastewater (200 ml, *n* = 19) and dH_2_O negative controls were clarified by centrifugation at 10,000× *g*, 4 °C, for 10 min. The supernatant (150 ml) was transferred to a sterile 250 ml PCCO bottle, and the pH was adjusted to 7–7.5 using 1 M NaOH. PEG-NaCI (50 ml) was then added to each sample, and the sample was mixed and incubated overnight at 4 °C. The following day, the samples in PEG-NaCI were centrifuged (10,000× *g*, 4 °C, 30 min). The resulting pellet was resuspended in 200 μl of PBS. All samples were processed in duplicate, one for RT-qPCR and one for ci-RT-qPCR. Concentrates were stored at 4 °C until nucleic acid extraction. All sample preparation and processing were conducted in a Biosafety Level 2 (BSL2) laboratory, adhering to WHO and national biosafety guidelines [50]. 

### 2.4. Nucleic Acid Extraction 

Nucleic acids were extracted from concentrates using NucliSens lysis buffer (BioMerieux, Marcy-lÉtoile, France, Cat No. 280134 or 200292), NucliSens extraction reagent kit (BioMerieux, Cat. No. 200293), and a Kingfisher 96 Flex system (Thermo Scientific, Waltham, MA, USA), following a previously published protocol [49]. Extracts were stored at −80 °C until analysis. 

### 2.5. Real-Time Quantitative PCR (RT-qPCR) and Capsid Integrity-RT-qPCR (ci-RT-qPCR) Analysis

All qPCRs (RT-qPCR and ci-RT-qPCR) were carried out on a Quant Studio Flex 6 Real-time PCR machine (Applied Biosystems Inc., Waltham, MA, USA). To quantify AdV, the qPCR reactions contained 2 X PowerUP™ SYBR™ Green Master Mix (Applied Biosystems™, ThermoFisher, Cat No: A25742), 10 pmol forward primer, 10 pmol reverse primer (Appendix A), molecular grade water, and a 4 µl sample, to make a 20 µl final reaction volume. The cycling conditions were to hold at 95 °C for 5 min, followed by 40 cycles of 95 °C for 0.15 s, 56 °C for 30 s annealing, and melt curve at 95 °C 15 s, then 60 °C for 1 min and 95 °C for 15 s at increments of 1.6° C/s. The qPCRs for EV, HAV, IAV, IBV, NoV GI, NoV GII, and SARS-CoV-2 contained 5 µl 4 × TaqMan™ Fast Virus 1-Step Master Mix for qPCR (Applied Biosystems™, ThermoFisher, Cat No: 4444432), 10 pmol forward primer, 20 pmol reverse primer, 5 pmol probe (Appendix A), 6 nmol MgSO_4_, 1 μg bovine serum albumin (BSA), molecular grade water, and a 4 µl sample to make a final reaction volume of 20 µl. RT-qPCR was conducted as follows: hold at 55 °C for 60 min for reverse transcription, 95 °C for 5 min for reverse transcriptase inactivation, followed by 45 amplification cycles of 95 °C for 15 s, 60 °C for 1 min, and 65 °C for 1 min at increments of 1.6 °C/s. All viral targets were run alongside a dsDNA (AdV), ssDNA (HAV), or ssRNA standard curve ranging from 1 to 10^5^ copies/µl The DNA standards were purchased from IDT (Integrated DNA Technologies Inc., Coralville, IA, USA), whilst the RNA standards were prepared in-house using a protocol previously described by Kevill et al. (2021) [49]. All samples were run in duplicate with each plate containing four non-template controls with molecular grade water. 

### 2.6. Data Analysis

The qPCR data were analysed and quality checked as per MIQE guidelines for RT-PCR [51]. The limit of detection (LOD) and limit of quantification (LOQ) for viral targets in wastewater have been descried previously [52]. The gene copies per PCR reaction were transformed into gc/ml for viruses in PBS and gc/l for viruses detected in wastewater, for statistical analysis. The data followed a non-normal distribution; therefore, nonparametric statistical analysis was performed using a Kruskal–Wallis test, followed by post hoc multi-comparisons using a Dunn’s test [53], with a Bonferroni corrected alpha of *p* < 0.003. Statistical analysis was performed using R studio, utilising the “MultNonParam” package, “Kruskal.test” [54], and “dunn.test” [55]. Log reductions were calculated as:(1)$$Log\;reduction=log10\;(\frac{Viral\;copies\;of\;untreated\;samples}{Viral\;copies\;of\;treated\;samples})$$

## 3. Results

### 3.1. PMAXX Dye Concentration on Live and Heat-Inactivated Viruses (AdV, HAV, IAV, NoV GI)

Comparisons of live and heat-inactivated viruses (HI) (AdV, HAV, IAV, and NoV GI) pretreated with PMAxx dyes at concentrations of 50 μM and 100 μM and untreated samples (no PMAxx dye) (Figure 1) were made using a Kruskal–Wallis test. The analysis revealed significant differences between all viral targets, across treatment groups (*p* < 0.001). The post hoc Dunn’s test with Bonferroni-corrected alpha of *p* < 0.003 identified significant differences between mean ranks of paired comparisons (Table 1). Pretreatment with the 100 μM PMAxx dye significantly reduced the AdV copy number compared to the untreated samples (no dye) (*p* 0.0024, Table 1). The HI AdV untreated had 4 × 10^7^ gc/ml versus 2 × 10^6^ gc/ml for live AdV, pretreated with 100 μM PMAxx dye (Figure 1). This result indicated the AdV sample contained degraded/semi-assembled virus. The HI 100 μM dye significantly reduced the viral copy number >100-fold compared to HI no dye for AdV and IAV (Table 1, Figure 1). A significant difference between the 100 μM pretreated live and 100 μM pretreated HI samples for NoV GI (*p* <0.0003, Table 1) also occurred, as the gc/ml were higher for live NoV GI (2 × 10^4^ gc/ml) than for the HI samples (5 × 10^3^ gc/ml) (Figure 1). The gc/ml was significantly lower for all viruses when comparing HI 100 μM to the live no dye treatment (Figure 1, Table 1), indicating the dyes at this concentration reduce the gene copies detected by RT-qPCR and are suitable for ci-RT-qPCR assays for the target viruses. 

### 3.2. Wastewater Samples 

Wastewater samples were negative for AdV, HAV, IAV, and IBV; so, a comparison of RT-qPCR and ci-RT-qPCR was conducted for EV, NoV GI, NoV GII, and SARS-CoV-2, as these viruses were detected in the samples. SARS-CoV-2 was detected in all wastewater samples via RT-qPCR only, while NoV GI, NoV GII, and EV were detected by both RT-qPCR and ci-RT-qPCR (Figure 2). EV was detected in 17 out of 19 samples, ranging from 7.82 × 10^2^ to 1.84 × 10^5^ and 6.14 × 10^2^ to 4.87 × 10^5^ gc/l for ci-RT-qPCR and RT-qPCR, respectively (Figure 2). Additionally, EV showed the lowest log reduction across all target viruses, at -0.52–1.15 (Figure 3), indicating that the EV capsid remains intact in wastewater. NoV GI and NoV GII were detected in all wastewater samples. NoV GI ranged from 5.89 × 10^3^ to 4.72 × 10^5^ and 1.43 × 10^3^ to 1.69 × 10^6^ gc/l for ci-RT-qPCR and RT-qPCR, respectively, and the log reductions were between 0.9 and 1.51 (Figure 3). NoV GII was detected at higher copy numbers than NoV GI, ranging from 9.20 × 10^2^ to 3.70 × 10^5^ and 3.72 × 10^4^ to 8.12 × 10^6^ gc/l, respectively (Figure 2). The log reductions for NoV GII were between 0.31 and 1.69 (Figure 3).

## 4. Discussion

Molecular methods used to assess the capsid integrity of viruses present in wastewater are needed to provide more in-depth risk analysis of human exposure to wastewater [38], especially where sewage is discharged into the environment, as it may contain intact potentially infectious viruses [56]. It may also be relevant for undertaking quantitative microbial risk assessments (QMRA) to estimate the potential viral exposure of operators at wastewater treatment plants or those working within the sewage network [57]. As evidenced, EV, NoV GI, and NoV GII capsids remain intact, and as a result are potentially infectious. 

In this study, we further advance the use of intercalating dyes for viral capsid integrity-RT-qPCR assays used in WBS applications. PMAxx dye at 50 μM and 100 μM concentrations was selected based on a review by Leifels et al. 2021 [38], where over half of the research papers reported dye concentrations of 50 μM and 100 μM to be sufficient for detecting damaged viral capsids. Due to the increased cost associated with using 200 μM of PMAxx dye, and the fact that 100 μM was deemed sufficient, we decided to assess lower concentrations of the dye. The heat-inactivated viruses exposed to PMAxx dye had a lower amplification resulting in lower gc/ml values than that of the live and no-dye-treated heat-inactivated viruses, a result consistent with other studies that assessed the effectiveness of PMAxx on heat treated Norovirus GI and GII and HAV [42,58]. In this study, we used heat inactivation to damage the viral capsid and compared the results to corresponding samples that did not undergo heat inactivation. Exposing viruses to high temperatures (>50 °C) causes them to become non-infectious and is a well-established method, as evidenced in cell culture assays [59]. However, using lower temperatures (<50 °C), while potentially inactivating viruses, may not sufficiently destabilise the viral capsid enough to expose the viral genome. In such cases ci-qPCR can overestimate the remaining infectious potential by detecting encapsulated viral genomes of viruses that can no longer replicate. Overall, the 100 μM dye concentration effectively differentiated live viruses from heat-inactivated viruses and reduced the detection of inactivated viruses for the targets tested. The PMAxx dyes also allowed us to identify viruses that were degraded or semi-assembled, as was the case for the AdV culture stock used in this study. These findings demonstrate that the PMAxx dye at 100 μM reduces the detection of heat-inactivated and degraded viruses compared to live viruses, allowing differentiation of capsid integrity by qPCR. Other studies have demonstrated that 50 µM PMAxx is sufficient for detecting human AdV in aquatic matrices [60] and similarly for NoV on vegetables [58]. However, the concentration of PMAxx dye treatment required varies depending on the sample type and what compounds maybe present in the sample [30]. Therefore, initial optimisation is required before the use of PMAxx in capsid integrity assays, across the sample type and target viruses.

Once the concentration of PMAxx dye, effective on several viruses, was determined, we tested our assay on wastewater samples for more targets (AdV, EV, HAV, IAV, IBV, NoV GI, NoV GII, and SARS-CoV-2). Wastewater samples were negative for AdV, HAV, IAV, and IBV. Therefore, we compared the RT-qPCR and ci-RT-qPCR data for EV, NoV GI, NoV GII, and SARS-CoV-2, which were detected in the wastewater samples. SARS-CoV-2 was detected in all samples by RT-qPCR only, showing that SARS-CoV-2 did not remain viable in the wastewater samples tested; as a result, the log reductions were not calculated, as no intact SARS-CoV-2 viruses were detected. The integrity of SARS-CoV-2 in wastewater samples has previously been investigated using PMAxx dyes and cell culture [31]. SARS-CoV-2 that had been spiked into wastewater had strongly (>3 log) or moderately (>1 log) reduced infectivity, and encapsulated SARS-CoV-2 RNA was always detected at a lower titre than total RNA in wastewater [31]. The reduced integrity of SARS-CoV-2 in wastewater is likely due to the enveloped capsid, which is less stable than that of non-enveloped viruses (e.g., NoVs, EV) [61], a result which mirrors the known modes of SARS-CoV-2 transmission. EV, NoV GI, and NoV GII were detected by RT-qPCR and ci-RT-qPCR. EV showed the lowest log reductions between RT-qPCR and ci-RT-qPCR, suggesting the viral particles remain intact and potentially infectious in wastewater. The EV RT-qPCR assay used in this study is a non-specific assay that detects a highly conserved region of the 62 nonpolio enteroviruses and three poliovirus types characterised at the time the assay was developed [62]. Therefore, the exact enterovirus strains detected are unknown. Norovirus GI and GII were detected in all samples, with NoV GII at higher gene copies. The log reductions for NoV GI and GII were < 1 log for most samples, indicating the presence of viable virus particles in the wastewater samples. PMAxx dyes have previously been used to detect viable NoV GI and GII in wastewater samples [30], while PMA dyes have been used to detect NoV GI and GII in contaminated drinking water [63], freshwater [29], and sewage samples [64]; again, these viruses remain intact and pose a human health risk. 

Reductions in viral gene copies for inactivated viruses were observed in our assays, and our results show that PMAxx dyes are suitable for use in WBS; however, there are limitations to the approach, and where possible, steps should be taken to avoid these. Intercalating dyes such as PMAxx rely on a photolysis step; therefore, the sample matrix needs to be considered, as shown when detecting viruses in shellfish [30], as when a sample has a high volume of suspended solids, light penetration maybe insufficient to induce the reaction. The floccing of viruses in the sample may also need to be considered, as damaged viruses in the middle of viral aggregations maybe protected from the dye, and samples may need to be agitated to limit this. In addition, the dyes cannot differentiate between non-viable viruses that have an intact capsid and live infectious viruses, as seen with UV treatment [36,39]. Furthermore, the target virus, concentration of dye, length of dye incubation period, and length of light exposure all need to be considered [38]. Therefore, while ci-RT-qPCR gives a better estimation of the number of viable virus particles than RT-qPCR alone, an overestimation of viable viruses within a sample may still occur. 

## 5. Conclusions

This study demonstrates the utility of PMAxx dye for quantifying viable viruses, even when cell culture-based assays are unavailable. With careful optimisation, ci-RT-qPCR can build upon knowledge from public health surveillance to provide estimates of viral infectivity. Assessing infectious viruses will enable better evaluation of potential public health risks associated with wastewater discharge and environmental transmission routes. Overall, capsid integrity-PCR shows promise for enhancing the specificity of wastewater-based epidemiology by revealing insights into community-level transmission dynamics of infectious pathogens.

## Figures and Tables

**Figure 1 viruses-16-00040-f001:**
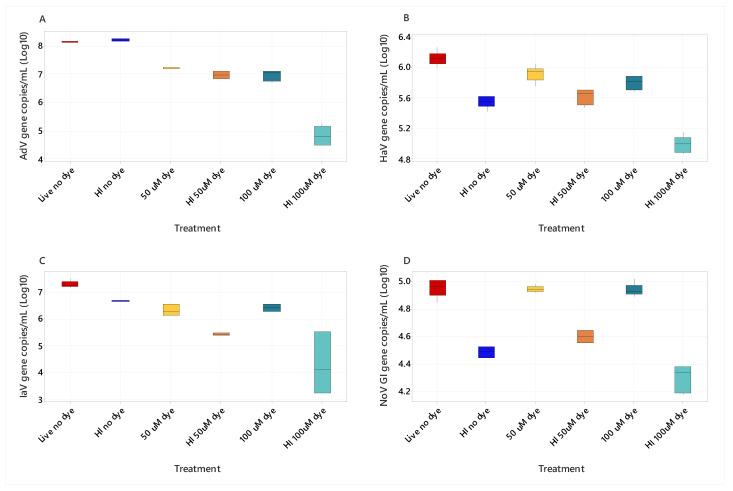
Effect of PMAxx addition (0, 50, or 100 µM) on the RT-qPCR-based quantification of live and heat-inactivated (HI) viruses (AdV (**A**), HAV (**B**), IAV (**C**), and NoV GI (**D**)).

**Figure 2 viruses-16-00040-f002:**
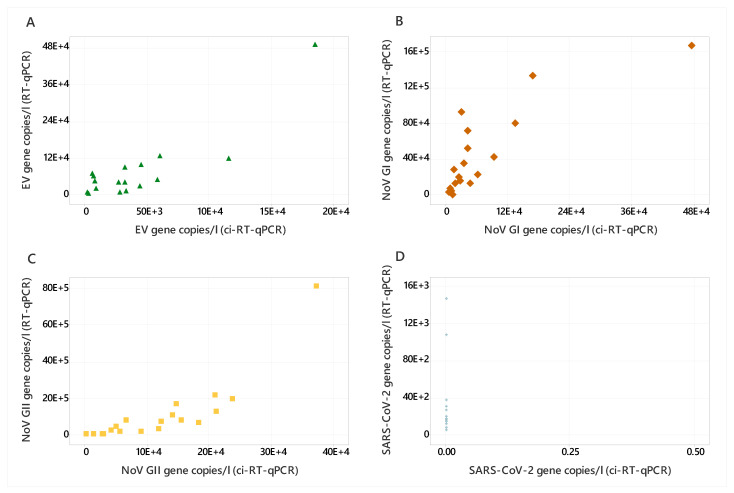
Scatter diagrams comparing RT-qPCR and ci-RT-qPCR gene copies/l for EV (**A**), NoV GI (**B**), NoV GII (**C**), and SARS-CoV-2 (**D**) detected in wastewater samples (n 19).

**Figure 3 viruses-16-00040-f003:**
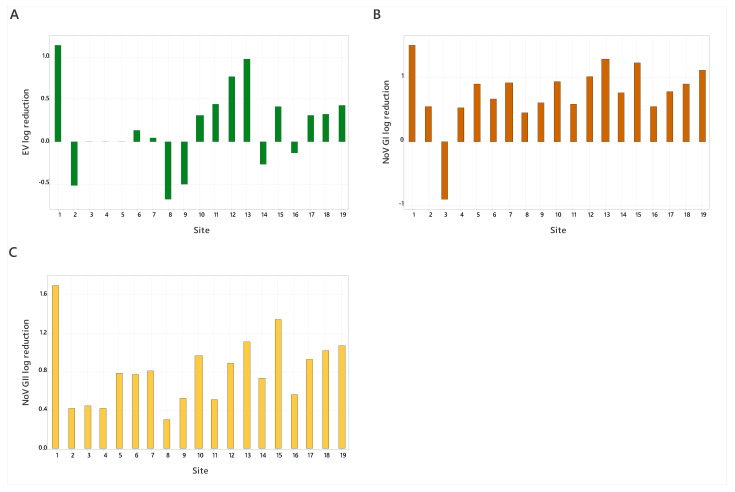
Log reductions of viruses (EV (**A**), NoV GI (**B**), and NoV GII (**C**)) detected in wastewater for samples 1–19.

**Table 1 viruses-16-00040-t001:** The *p* values from Dunn’s multi-comparison post hoc test, showing the significance between ranked pairs for each target virus. Comparisons were made between the gene copies/ml obtained from live and heat-inactivated (HI) viruses (AdV, HAV, IAV, or NoV GI) that were exposed to 50 µM or 100 µM PMAxx dye or no dye. Significant (Bonferroni corrected) *p* values are highlighted in bold.

Comparison	AdV	HAV	IAV	Nov GI
Live 100 µM—Live 50 µM	0.2700	0.4300	0.6800	0.8813
Live 100 µM—HI 100 µM	0.2500	0.0037	0.0260	**0.0003**
Live 100 µM—HI 50 µM	0.8800	0.3200	0.0820	0.1419
Live 100 µM—HI no dye	**0.0024**	0.0870	0.1700	0.0112
Live 100 µM—Live no dye	0.0250	0.0520	0.0160	0.5462
Live 50 µM—HI 100 µM	0.0140	**0.0002**	0.0680	**0.0002**
Live 50 µM—HI 50 µM	0.2000	0.0890	0.1700	0.1091
Live 50 µM—HI no dye	0.0310	0.0130	0.0770	0.0072
Live 50 µM—Live no dye	0.2000	0.2500	0.0049	0.6447
HI 100 µM—HI 50 µM	0.3300	0.1100	0.8000	0.0780
HI 100 µM—HI no dye	**<0.0001**	0.2300	**0.0003**	0.2825
HI 100 µM—Live no dye	**0.0003**	**0.0000**	**<0.0001**	**0.0001**
HI 50 µM—HI no dye	**0.0013**	0.6000	**0.0031**	0.4231
HI 50 µM—Live no dye	0.0160	0.0062	**0.0001**	0.0502
HI no dye—Live no dye	0.3700	**0.0003**	0.3000	**0.0025**

## Data Availability

Data are contained within the article or Appendix A.

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
