# Peer review of "Use of Capsid Integrity-qPCR for Detecting Viral Capsid Integrity in Wastewater"

_viruses, 2023, doi:10.3390/v16010040_

Round 1

Reviewer 1 Report

Comments and Suggestions for Authors

Thank you for submitting your very sound and solid investigation into the utilization of azo dyes to investigate virus infectivity in environmental samples (a topic that is very dear to me).

I read the manuscript with great interest and have only a few minor comments and recommendations before it can be accepted for publication. The main point might be the easiest to address: you refer to the quantitative PCR after azo-dye pretreatment as viability qPCR which I think is wrong as viruses do not technically live (no cellular activity, no intake/excretion of nutrients, no reproduction etc) and therefore have no “viability” to speak of. We have coined the term “capsid-integrity (ci-) qPCR” in the past to ensure a bit more precision regarding the use of dyes for viruses. Please consider replacing viability qPCR (which is totally fine for bacteria, by the way) and to use ci-qPCR instead.

Further comments and edits can be found in the manuscript itself.

Author Response

Thank you for your kind review! We have loved using the days to assess capsid integrity in wastewater, we are pleased you enjoyed reading our paper.

Thank you for your suggestion, we have changed the word viability throughout manuscript.

Abstract – Yes, it is true that some UV light does not damage a viral capsid enough to unencapsulated the genome inside, we have addressed this in the introduction and discussion. We are limited for words in the abstract, and do not feel it necessary to raise this point here.

Line 36 – We have changed to WBS.

Line 39 – Updated the text to say non-polio and polio enteroviruses.

Line 43 – removed the word outbreak.

Line 60 – Thank you very much, we have added this to the MS. We also subscripted ‘2’.

Line 79 – We had added some more text, which will allow readers to understand the difference between damaged vial capsids and intact viruses.

Line 84 – Thank you for pointing this out, we have amended the sentence slightly. We feel that the wording is careful as we use ‘potentially infectious’ , we are not suggesting that viruses are inherently infectious. For clarity we have made changes to the language throughout the MS.

Line 70 – We used the heat inactivated viruses as a control to ensure the PMAxx was working effectively.  Added to text and added to the discussion (line 274-281).

Line 126 – Thank you for your question. We did not test a lower concentration of PMAxx, as most research is conducted using a higher concentration of dye. For our samples and viral targets, a higher concentration of dye worked better, so we did not even think to try anything lower than 50 um.

Line 164 – This SYBR assay is optimised for use in wastewater and is the one we use in our lab for AdV surveillance, no other reason than that.

Figure 1 – Changed to box plots.

Figure 2 –  The qPCR and ci-qPCR data is for each corresponding sample, we would have to plot the error for each method (2) on one sample point, this is not possible for this type of plot. We simply want to show the corresponding copy number determined by qPCR and ci-qPCR for each wastewater sample.

Line 249 – Thank you for this information, I have added the term QMRA.

Line 318-319 – we now raise this in the introduction too.

Reviewer 2 Report

Comments and Suggestions for Authors

This paper concerns the use of nucleic acid-binding dye to distinguish between potentially infective and non-infective viruses in wastewater. It was interesting to learn more about this topic. I found the 2010 paper by Parshionikar et al. in Appl. Environ, Microbiol. to be very instructive. Given that this manuscript resembles that paper in many ways, I was surprised that it was not cited. While this type of research has apparently been going on for the last 10 years or so, as the authors point out, there has not been a lot of it, which makes this contribution of some value. 

I have a few questions and suggestions for improvement.
Heat treatment is a convenient method of viral inactivation, but other mechanisms (UV exposure, chlorination) are more relevant to a wastewater environment. Other research has used these treatments for viral inactivation. Wouldn't the results be better supported if an additional viral inactivation treatment was used?
Methods: As I understand it, PMA readily binds to ds DNA. If PMA isn't washed away prior to nucleic acid extraction, doesn't that create an opportunity for it to bind to DNA if the virus being detected is a ds DNA virus (e.g. adenovirus, used in this study)?

Fig. 2 The axis labels are confusing. A slash mark is usually read as "per". Should these axis labels be translated as "gene copies as determined from RT-qPCR" vs. "gene copies as determined from V-RT-qPCR (with PMAxxx)"? I would suggest a clarifying edit. Although it is not possible to standardize all the axis ranges, it would help to make more of them the same for easier reading. Also, hard to read all those zeroes in Fig. 2 A-C. Please write these in scientific notation for easier reading.
In this manuscript, viruses are treated to make them non-infectious, but there is no validation or quantification shown of non-infectivity. The similar paper by Parshionikar et al. used actual infectivity assays. Since the use of the PMA dye is what is being tested in this manuscript, it cannot also be used as an independent measure of the extent of viral inactivation. This needs to be addressed in some fashion, either with additional data or with explicit references showing the expected degree of inactivation under identical conditions.
There is a problem with Fig. 3. In Fig. 2D, the graph shows that there are no gene copies recovered from SARS-CoV2 when measured using V-RT-qPCR. However, the way the Methods section defines Log Reduction, it appears that the denominator for SARS-CoV2 should therefore be 0, the Log Reduction value would be impossible to calculate, and Fig. 3D should be ungraphable. Please correct or explain this.

Author Response

Thank you for your review and questions. The Parshionkar paper is interesting, and we have now cited it in the paper. Thank you for this.

Thank you for the suggestion. The heat treatment used in the study, was to act as control for testing the PMAxx dyes on viral targets in PBS, not to mimic heat treatment in wastewater treatment processes. We have added more text to the methods to clarify this point (line 122). The purpose of this paper is the assess viral capsid integrity using PMAxx dyes in untreated wastewater, as untreated wastewater poses a larger health risk to the public and is regularly discharged into bathing waters in the UK. This is stated in the introduction (line 54) and then again in the discussion (line 257). The purpose of the paper is to show that this method works for detecting encapsulated viruses in UK wastewater, it can then be applied to other studies. I appreciate that previous research covers water treatments for the inactivation of viruses, but this is outside of the scope of this paper. 

Thank you for this, PMAxx dyes require the photolysis step to bind to nucleic acids, after photolysis the dyes can longer bind to DNA. Therefore, not an issue for downstream methods.

Fig.2. The axis labels are gene copies/l qPCR or ci-RT-qPCR. For clarity we have changed them to gene copies/l (qPCR) or gene copies/l (ci-RT-qPCR), as I think the reviewer found the axis hard to read.

Thank you for this – Heat-inactivation of viruses is a well-established method, and there is a wealth of research that shows how effective heat-inactivation is at denaturing viral capsids. We use the heat inactivated virus as a control to show the PMAxx dyes are working, as we do not have cell culture facilities or a BSL 3 lab to culture viruses in wastewater. We have addressed your comment in the discussion and cite a review paper that evaluates the thermal inactivation of viruses, which also covered the viruses we use and the heat inactivation method.

Figure 3 –We calculate this by taking the copy number of the qPCR data and deducting this for the copy number of  ci-qPCR data for the same sample. In the case of SARS the log reduction between qPCR and ci-qPCR is actually the log10 of what we detected by qPCR, as we did not detect any intact virus via CI-qPCR. The figure is not incorrect, it just shows the difference between the data. We had added a sentence line 300 in the discussion for clarity.

Round 2

Reviewer 1 Report

Comments and Suggestions for Authors

thank you for your very kind reply and for the detailled responses to all the remarks and comments I made.

I fully recommend accepting the manuscript in its current form and to proceed with the publication process.

Author Response

Thank you so much!

Reviewer 2 Report

Comments and Suggestions for Authors

The updated version of the manuscript is almost ready for publication. I should note that there are a few minor typos scattered through the document that could be corrected following an additional proofreading. I only have two lingering concerns related to the figures.
I assume that the original version of Fig. 2 was left for comparison and will be removed in the final version.
I still do not undrestand the explanation for Fig. 3 D. The Y axis says Log Reduction. The Method section states that this is calculated as Log10 of {gene copies determined by qPCR} divided by {gene copies determined by ci-qPCR}. Fig. 2 shows that the gene copies for SARS CoV2 determined by ci-qPCR were 0. One cannot divide by 0. Unfortunately, there is no correct mathematical way to calculate Log Reduction for SARS CoV2. Figure 3D cannot exist.

Author Response

Thank you for your review! 

The final version of the MS has been proof read, and typos amended. 

Apologies, I have no idea what happened to fig 2. This has now been corrected. 

Fig 3 - we have removed D and amended the legend for clarity.